# Memory-Augmented Large Language Model-Based Agent with Cross-Task Experience Learning

## Abstract

Large Language Model (LLM)-based agents have demonstrated impressive capabilities in complex decision-making and multi-turn instruction-following tasks. To enhance knowledge retention and contextual adaptability, recent work has equipped these agents with memory modules that store and reuse historical interaction experiences. However, existing memory-augmented approaches face two key limitations: they often require large amounts of interaction data during early training to reach competitive performance, resulting in low data efficiency; and they rely on static, self-derived experience reuse strategies, limiting their ability to adapt when prior learning is insufficient and preventing the use of transferable knowledge from related tasks. Building on these observations, in this paper, we propose a memory-augmented LLM agent with cross-task experience learning, designed to improve data efficiency and adaptability. Our method augments the conventional task-specific memory with an additional source experience memory that retains transferable knowledge from related but distinct tasks. We further introduce a dynamic memory retrieval mechanism that adaptively draws from both task and source memories, allowing the agent to balance prior task-specific experiences with cross-task knowledge according to the current context and progression. We validate the proposed method on the WebShop benchmark, which comprises diverse, multi-turn instruction-following tasks across product domains with varying semantic complexity. Experimental results show that our approach consistently outperforms state-of-the-art memory-augmented LLM agents in task success rate and generalization, demonstrating the effectiveness of the proposed memory architecture and retrieval mechanism.

## 1 Introduction

With the advancement of Transformer architectures and large-scale pre-training techniques, large language models (LLMs) have demonstrated remarkable capabilities in natural language understanding Shi et al. (2024), reasoning Huang & Chang (2023); Singh et al. (2023), and generation Stiennon et al. (2020); Bian et al. (2024); Akyürek et al. (2023). Building upon these strengths, LLMs are increasingly established as a core component of autonomous agents, with a growing number of studies leveraging them as the foundation for high-level decision-making Chen et al. (2019); Shinn et al. (2023); Shen et al. (2023); Zhu et al. (2023b); Sclar et al. (2023). These agents have demonstrated promising performance across diverse tasks such as decision planning Karpas et al. (2022); Du et al. (2023), information retrieval Nakano et al. (2022); Ruan et al. (2023), and tool use Schick et al. (2023). However, most existing LLM-based agents treat each interaction as an isolated event, lacking effective mechanisms to incrementally accumulate and reuse task-specific knowledge. This limitation severely impairs their ability to adapt strategies based on prior experience, reducing data efficiency and adaptability in continuously evolving task environments.

To address the limitation of LLM-based agents in retaining and leveraging prior experience for strategy adaptation, recent research has increasingly explored augmenting LLM-based agents with memory modules designed to support knowledge retention and contextual adaptation throughout multi-turn interactions Rana et al. (2023); Zhu et al. (2023a); Wang et al. (2025); Huang et al. (2023b). By incorporating mechanisms that emulate human memory processes, these approaches

aim to facilitate the gradual accumulation of task-specific knowledge and enhance the agent's ability to generalize across dynamic and evolving task environments. For instance, Reflective Linguistic Programming (RLP) embeds the agent's recent observations into prompts to simulate short-term memory Fischer (2023). Generative agents extend this idea by combining short-term and long-term memory structures Park et al. (2023). Reflexion introduces feedback-driven self-reflection that integrates memory for iterative strategy refinement Shinn et al. (2023). More recently, REMEMBERER frames the LLM as a reinforcement learning agent and augments it with an experience memory module, representing a more advanced approach for retaining and retrieving interaction histories Zhang et al. (2023). While memory-augmented designs improve adaptability, they still face two fundamental limitations. First, they typically rely solely on self-derived experience, lacking mechanisms to leverage transferable knowledge from related tasks. As a result, they require extensive interaction data in early training to compensate for knowledge gaps, leading to low data efficiency. Second, most existing methods adopt static, manually designed memory retrieval strategies, which constrain the agent's ability to adaptively reuse experiences based on task context and progression.

Building on these observations, in this paper, we propose a memory-augmented LLM-based agent with cross-task experience learning, aimed at improving both data efficiency and adaptability. In particular, our method augments the conventional task-specific memory with an additional source experience memory that retains transferable knowledge from related but distinct tasks, enabling more efficient reuse of prior learning. Meanwhile, the task-specific memory itself is enhanced to more effectively capture context-dependent information during ongoing interactions, thereby supporting fine-grained adaptation to the current task. To effectively leverage these two forms of memory, we further introduce a dynamic retrieval mechanism that adaptively balances task-specific experiences with cross-task knowledge according to the current interaction state and progression, thereby enhancing both decision quality and generalization across tasks. We evaluate our method on the WebShop benchmark Yao et al. (2022), which involves diverse, multi-turn instruction-following tasks spanning product domains with varying semantic contexts. Experimental results show that our approach consistently achieves higher task success rates and better generalization compared to state-of-the-art memory-augmented LLM-based agents, validating the effectiveness of the proposed memory architecture and dynamic retrieval mechanism.

Our main contributions are summarized as follows: (1) We propose a memory-augmented LLM-based agent with cross-task experience learning, which improves data efficiency and adaptability by enabling the reuse of transferable knowledge from related but distinct tasks; (2) We design a dynamic memory retrieval mechanism that adaptively balances task-specific and cross-task experiences according to the agent's interaction state and task progression, thereby supporting more effective decision-making; (3) We conduct extensive experiments on the WebShop benchmark, demonstrating that our method consistently outperforms state-of-the-art memory-augmented LLM-based agents in both task success rate and generalization across diverse task domains.

## 2 RELATED WORK

This section reviews the studies relevant to this work, including LLM-based agents and memory-augmented LLM-based agents.

**LLM-Based Agents** Large language models (LLMs) have shown strong abilities in language understanding, reasoning, and decision-making, enabling their integration as core components in autonomous agents for complex environments. Early approaches primarily relied on prompting for action generation. For instance, SayCan Ahn et al. (2022) couples LLMs with an affordance model for robotic control but lacks explicit feedback and memory for long-term planning. Inner Monologue Huang et al. (2023a) incorporates environment-informed feedback for iterative refinement, yet depends on predefined signals without mechanisms to accumulate knowledge. WebGPT Nakano et al. (2022) enables browsing and information retrieval, but relies heavily on costly human feedback and lacks explicit reasoning modeling. ReAct Yao et al. (2023) interleaves reasoning and action to produce interpretable traces, improving task performance but still treating each interaction as isolated. Despite these advances, most LLM-based agents remain limited in long-term, consistent, and adaptive decision-making. By failing to incrementally accumulate and reuse prior experiences, they struggle to adapt strategies over time, resulting in low data efficiency and brittle performance in dynamic, multi-turn environments.

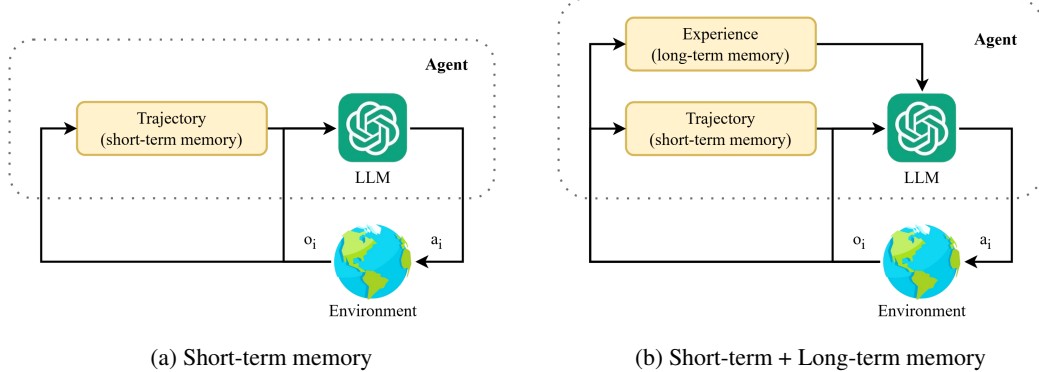

(a) Short-term memory        (b) Short-term + Long-term memory

Figure 1: Architectures of LLM-based agents with different memory mechanisms.

**Memory-augmented LLM-based Agents** To address the limited capacity of LLM-based agents to accumulate and reuse experience, recent works introduce memory modules for knowledge retention and contextual adaptation across interactions. As shown in Fig. 1a, early efforts focus on short-term memory, leveraging only the current trajectory. For example, Reflective Linguistic Programming (RLP) Fischer (2023) injects recent observations directly into prompts, but lacks mechanisms to summarize or reuse past episodes, limiting long-horizon planning. DEPS Wang et al. (2023) and Corrective Re-Prompting Raman et al. (2022) utilize single-step corrective signals to refine responses. While effective at resolving local errors, they cannot accumulate or generalize knowledge across tasks. Consequently, such approaches treat each episode as isolated, preventing systematic knowledge reuse. To overcome these issues, more recent work explores long-term memory in combination with short-term modules (Fig. 1b). ChatDB Hu et al. (2023) and Ret-LLM Modarressi et al. (2024) employ external storage to persist environment states or feedback, but these modules are loosely coupled with task reasoning and updated infrequently, hindering responsiveness. Reflexion Shinn et al. (2023) integrates short- and long-term memory with feedback-driven self-reflection, yet its long-term store is severely capacity-limited, retaining only a small subset of experiences. GITM Zhu et al. (2023b) buffers successful trajectories, but its unstructured memory format impedes efficient retrieval and abstraction, restricting scalability. REMEMBERER Zhang et al. (2023) augments reinforcement learning with extended long-term memory for adaptive decision-making, but requires extensive interaction data and relies on fixed retrieval rules, limiting adaptability under shifting task demands.

Although memory-augmented agents have advanced beyond purely prompt-based approaches, existing methods still face critical limitations. First, they typically require large amounts of interaction data during early training to populate memory, resulting in low data efficiency. Second, most approaches rely on static, manually designed memory retrieval strategies, which restrict the flexibility and responsiveness of memory utilization. Consequently, these methods primarily exploit an agent's own accumulated experiences, overlooking the potential benefits of cross-task knowledge transfer. Such transfer, however, holds considerable promise for improving data efficiency, particularly when the current task bears significant similarity to previously encountered ones.

## 3   PROPOSED METHOD

In this section, we present a detailed description of the proposed memory-augmented LLM-based agent with cross-task experience learning. We first outline the overall architecture, then elaborate on the target memory update process, and finally describe the dynamic retrieval mechanism.

### 3.1   ALGORITHM OVERVIEW

As illustrated in Figure 2, we propose a memory-augmented framework for LLM-based agents with cross-task experience learning. Our method extends conventional task-specific memory with an additional *source experience memory* that retains transferable knowledge from related but distinct

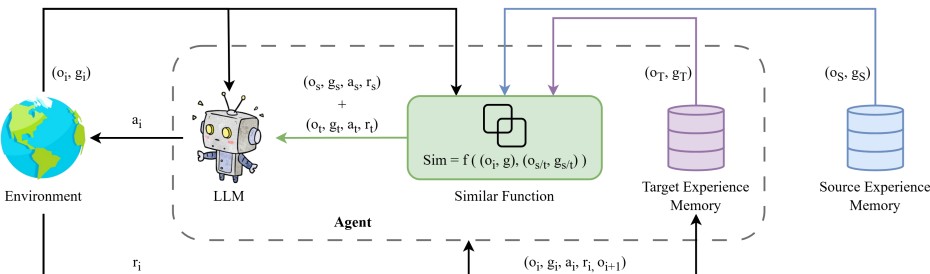

Figure 2: Architecture of the proposed memory-augmented LLM agent with cross-task experience learning

tasks. The agent dynamically leverages both memories to enhance decision-making across diverse tasks.

The *target experience memory* $\mathcal{M}_t$ incrementally accumulates task-specific experiences during on-going interactions, enabling continual adaptation to the current task. Specifically, it stores experience tuples $(o_t, g_t, a_t, r_t)$, where $o_t$ is the current observation, $g_t$ the task goal, $a_t$ the action taken, and $r_t$ the resulting reward. The *source experience memory* $\mathcal{M}_s$ retains high-utility experiences from previously completed tasks, facilitating cross-task generalization and reducing the need for extensive task-specific exploration. Each entry in $\mathcal{M}_s$ is an experience tuple $(o_s, g_s, a_s, r_s)$, where $o_s$ is a past observation, $g_s$ the associated goal, $a_s$ the action executed in that context, and $r_s$ the reward received. Only successful or high-value experiences are included to minimize noise and maximize transfer effectiveness.

At each decision step $i$, the agent observes the current state and task goal $(o_i, g_i)$ and retrieves relevant experiences from both $\mathcal{M}_s$ and $\mathcal{M}_t$ using a similarity-based retrieval function. Retrieval is dynamically weighted: early in an episode, both memories contribute to leverage transferable knowledge, while as $\mathcal{M}_t$ accumulates sufficient experiences, retrieval gradually prioritizes target memory to mitigate negative transfer. The number of retrieved entries is adaptively adjusted to fit within the LLM's token budget, ensuring maximal informativeness.

Retrieved experiences are converted into a structured natural language template capturing observation, goal, action, and reward (Figure 3). These are concatenated with the current observation and goal $(o_i, g_i)$ to form a context-enriched prompt for the LLM, which then generates the next action $a_i$. After execution, the resulting transition $(o_i, g_i, a_i, r_i, o_{i+1})$ is appended to $\mathcal{M}_t$, enabling continual task-specific adaptation. Optionally, Q-values for state-action pairs are maintained to support similarity-weighted retrieval and ongoing policy refinement.

By structuring the agent around source and target memories with a dynamic retrieval mechanism, the proposed method efficiently leverages both transferable and task-specific experiences. This design improves decision quality, reduces the need for extensive task-specific interactions, and enhances generalization across diverse multi-turn tasks.

## 3.2 SOURCE MEMORY CONSTRUCTION AND TARGET EXPERIENCE UPDATE

To support cross-task knowledge transfer, we pre-construct a *source experience memory* $\mathcal{M}_s$ that encapsulates transferable knowledge distilled from previously completed tasks. Each source experience is represented as a tuple

$$(o_s, g_s, a_s, r_s),$$

where $o_s$ denotes the environment observation, $g_s$ the task instruction or goal, $a_s$ the executed action, and $r_s$ the received reward. These experiences are collected from diverse source tasks to capture general strategies and patterns that are likely to benefit future tasks. Unlike the target memory, $\mathcal{M}_s$ is *static* during task execution—it is not updated with new interactions—providing a stable prior that informs the agent's decisions and facilitates cross-task generalization. The experiences in $\mathcal{M}_s$ are encoded in a structured, language-readable format compatible with prompt construction, allowing them to be retrieved efficiently by the dynamic memory retrieval mechanism described in Section 3.3. This design ensures that the agent can leverage prior knowledge from related tasks,

```
Last 5 Actions:
- search[high speed hdmi cables 10 feet 5 pack] ...
- click[b08qsnm69h] b08qsnm69h is a 10 feet ...
- click[5-pack] The item meets most of the ...

Observation:
You have clicked 5-pack...
Instruction:  i need high speed hdmi cables that ...
[Back to Search]
size:  [1-pack][3-pack][5-pack][10-pack] ...
color:  [black braided][blue braided] ...
4K HDMI Cables, Short Cord, 1ft [3 Pack] Braided ...
Price:  $15.xx
[Features]
...
[Buy Now]

Encouraged:
click[Buy Now] -> 1.0 The item meets all the requirements and is within the
desired price range.  You can proceed to purchase it.

Discouraged:
click[Back to Search] -> 0.0 The current item offers the desired options
and I don't need to search for other items.
```

Figure 3: An example of the memory format used in the WebShop Yao et al. (2022) task. Each retrieved experience tuple $(o_{s/t}, g_{s/t}, a_{s/t}, r_{s/t})$ is transformed into a structured textual representation using a predefined template.

reducing the need for extensive exploration in the target task and improving data efficiency from the outset.

To enable continual adaptation within the current task, the *target experience memory* $\mathcal{M}_t$ is incrementally updated with new trajectories collected during interaction. In contrast, the *source memory* $\mathcal{M}_s$ remains fixed across tasks, serving as a stable prior distilled from previously completed tasks. This separation ensures that $\mathcal{M}_s$ encodes transferable knowledge, while $\mathcal{M}_t$ focuses on capturing task-specific regularities.

At each decision step $i$, given the environment state $o_i$ and task instruction $g_i$, the agent executes an action $a_i$, receives a reward $r_i$, and transitions to $o_{i+1}$. The transition $(o_i, g_i, a_i, r_i, o_{i+1})$ is appended to $\mathcal{M}_t$, progressively enriching the task-specific knowledge base.

Beyond storage, $\mathcal{M}_t$ is continuously refined through reinforcement learning (RL), which estimates the long-term utility of experiences and guides policy improvement. Specifically, a Q-value function $Q(g, o, a)$ is maintained and updated using the Bellman target:

$$Q'(g, o_i, a_i) = r_i + \gamma \max_a Q(g, o_{i+1}, a), \tag{1}$$

where $\gamma$ is the discount factor. For tasks with large or unbounded action spaces (e.g., natural language), the $\max_a$ operator is approximated over a subset of observed actions. The Q-function is then updated using the standard Q-learning rule Watkins & Dayan (1992):

$$Q(g, o_i, a_i) \leftarrow (1 - \alpha)Q(g, o_i, a_i) + \alpha Q'(g, o_i, a_i), \tag{2}$$

where $\alpha$ is the learning rate.

The enhanced $\mathcal{M}_t$ thus fulfills two complementary roles: (i) providing task-specific experiences that can be retrieved and encoded into LLM prompts for context-aware decision-making, and (ii) supporting policy refinement through Q-value updates, enabling the agent to adapt more effectively over time.

### 3.3 DYNAMIC MEMORY RETRIEVAL MECHANISM

While the source memory $\mathcal{M}_s$ provides transferable priors and the target memory $\mathcal{M}_t$ accumulates task-specific knowledge, their utility depends critically on how relevant experiences are retrieved.

To this end, we propose a *Dynamic Memory Retrieval Mechanism (DMRM)* that adaptively balances cross-task transfer and task-specific specialization.

At each decision step $i$, given the current observation–goal pair $(o_i, g_i)$, the agent ranks candidate entries $(o_{s/t}, g_{s/t}, a_{s/t}, r_{s/t})$ from $\mathcal{M}_s \cup \mathcal{M}_t$ using contextual similarity:

$$Sim = f\big((o_i, g_i), (o_{s/t}, g_{s/t})\big), \tag{3}$$

where $f(\cdot)$ is the cosine similarity between sentence embeddings. The top-$k$ most relevant experiences are selected to guide inference.

To satisfy the input constraint of the underlying LLM, the number of retrieved entries $k$ is dynamically adjusted. Let $\text{len}(\cdot)$ denote token length and $C_{\max}$ the model's context window. Retrieval proceeds until:

$$\sum_{j=1}^{k} \text{len}(m_j) + \text{len}(o_i, g_i) \leq C_{\max}, \tag{4}$$

ensuring that retrieval remains length-aware and maximally informative.

Beyond relevance and length, DMRM introduces an *adaptive source control* strategy to mitigate negative transfer. Specifically, once $\mathcal{M}_t$ contains at least $n$ high-quality successful trajectories, retrieval from $\mathcal{M}_s$ is reduced or disabled:

$$|T_s| \geq n \quad \Rightarrow \quad \text{disable source retrieval},$$

where $T_s$ is the set of successful trajectories in $\mathcal{M}_t$ and $n$ is a configurable threshold. This mechanism allows the agent to rely on transferable priors in the early stage of adaptation, while gradually shifting to task-specific experiences as evidence accumulates.

Finally, the retrieved experiences are serialized into natural language using a unified template (Figure 3), concatenated with the current $(o_i, g_i)$, and passed to the LLM for action generation. This retrieval process ensures that decision-making is both context-aware and adaptively balanced between cross-task knowledge transfer and in-task learning.

## 4 EXPERIMENT

In this section, we empirically evaluate the proposed method on the WebShop benchmark, comparing its performance with standard RL baselines and state-of-the-art LLM-based agents.

### 4.1 EXPERIMENTAL SETUPS

#### 4.1.1 EVALUATION ENVIRONMENT

In this study, the WebShop Yao et al. (2022) serves as the experimental environment, offering a realistic simulation of multi-step online shopping via a web interface. It comprises over 1,000,000 product descriptions and 12,000 human-written instructions, supporting rigorous evaluation of instruction-following agents. In each task, the agent receives a user instruction and interacts with structured web pages via a discrete action set to identify a matching product. Only the five most recent interaction steps are visible, increasing the challenge of long-horizon reasoning. Episodes terminate upon product selection or after 15 steps, with a scalar reward in $[0, 1]$ reflecting task success.

WebShop spans diverse domains (e.g., beauty, electronics, grocery), with consistent UI structure but varying semantics and reasoning demands. We treat each domain as a separate task and assess cross-domain generalization by designating one as the source and others as targets. Following REMEMBERER Zhang et al. (2023), we adopt its setup: training on 10 randomly sampled tasks (disjoint from test tasks) for 3 epochs, repeated over five source–target pairs.

Memory retrieval employs a two-level similarity metric. Instruction-level similarity ($f_g$) is computed using SentenceTransformer all-MiniLM-L12-v2 Reimers & Gurevych (2019), while observation-level similarity ($f_o$) leverages the fixed layout structure of WebShop pages, assigning each to one of four predefined types via a lookup table. The final retrieval score is computed as in Equation (3).

### 4.1.2 COMPARED METHODS

To evaluate the effectiveness of our method, we compare it against several baselines. First, we consider an **LLM-only** setting, where a static GPT-3.5[1] or GPT-4[2] model is used with two fixed exemplars, without memory augmentation or task adaptation. We also include **ReAct** Yao et al. (2023), a strong baseline that interleaves reasoning and actions but relies solely on current observations and instructions, without leveraging past experiences. In addition, we compare with **RE-MEMBERER** Zhang et al. (2023), an LLM-based agent with memory-augmented RL that retrieves two fixed exemplars from its experience memory but lacks both cross-task knowledge transfer and adaptive memory selection during execution. Finally, we report results for standard **RL**, **IL**, and **IL+RL** agents, following Yao et al. (2023). To ensure a fair comparison, we align most hyperparameters with REMEMBERER Zhang et al. (2023). Unlike REMEMBERER, our method employs (1) a source memory storing transferable experiences and (2) a dynamic retrieval mechanism that selects relevant cases based on the current state and task progression, enabling more adaptive and context-aware memory utilization.

### 4.1.3 EVALUATION METRICS

We evaluate the effectiveness of our memory-augmented LLM agent with cross-task experience learning on the WebShop benchmark using two standard metrics. The **average score** is a scalar feedback in the range $[0, 1]$, where higher values indicate better alignment between the selected product and the user instruction. The **success rate**, ranging from $[0, 1]$, indicates the all task completion, with each task counted as a success (1) or failure (0). In both metrics, higher values indicate better agent performance. These metrics are widely adopted in prior works on instruction-following agents Yao et al. (2023); Zhang et al. (2023) because they effectively capture both qualitative and quantitative aspects of agent performance, balancing reward precision and task completion robustness.

### 4.2 RESULTS AND DISCUSSIONS

All agents in our experiments are powered by an instruction-tuned LLM that remains frozen throughout evaluation, without any gradient-based fine-tuning. Instead, their decision-making relies on carefully designed prompt engineering to elicit reasoning and adaptation behaviors.

Table 1: Overall performance comparison across baseline and proposed methods on the WebShop benchmark. Metrics include average instruction satisfaction score and task success rate.

| Method | Avg Score | Success Rate |
|---|---|---|
| LLM only | 0.55 | 0.29 |
| ReAct | 0.66 | 0.36 |
| RMMBR. (w/o transfer) | 0.68 | 0.38 |
| Ours (Mem+Transfer) | **0.70** | **0.40** |
| RL | 0.55 | 0.18 |
| IL | 0.60 | 0.29 |
| IL+RL | 0.62 | 0.29 |

Table 1 reports performance on the target category in terms of average score and success rate. Our method with cross-task transfer consistently outperforms all baselines, including LLM-only prompting, ReAct Yao et al. (2023), REMEMBERER without transfer Zhang et al. (2023), and standard RL, IL, and IL+RL approaches. It achieves the highest average score (0.70) and success rate (0.40), underscoring the benefits of incorporating transferable experience into LLM-based agents. These gains can be attributed to two key components of our design. First, the source experience memory provides access to transferable knowledge from related tasks, reducing the reliance on large amounts of task-specific interaction data and thereby improving data efficiency. Second, the dynamic memory retrieval mechanism adaptively balances source and target experiences, allowing the agent to

---

[1] https://platform.openai.com/docs/models/gpt-3.5-turbo
[2] https://platform.openai.com/docs/models/gpt-4-turbo

exploit transferable knowledge when beneficial while specializing on target-specific details as interaction progresses. Together, these mechanisms enable the agent to generalize more effectively across domains, yielding robust performance even under limited prior task experience.

Table 2: Transfer performance under **different target tasks**. For each method, the first row reports the mean scores, and the second row reports the success rates. Task abbreviations: **fa** = fashion, **be** = beauty, **el** = electronics, **gr** = grocery.

| Method | $\mathcal{M}_{t_{be}}$ | $\mathcal{M}_{t_{el}}$ | $\mathcal{M}_{t_{gr}}$ | Avg |
|---|---|---|---|---|
| LLM only | 0.55 | 0.53 | 0.57 | 0.54 |
| LLM only | 0.28 | 0.26 | 0.30 | 0.28 |
| RMMBR. (w/o transfer) | 0.68 | 0.66 | 0.70 | 0.68 |
| RMMBR. (w/o transfer) | 0.38 | 0.37 | 0.39 | 0.38 |
| | $\mathcal{M}_{s_{fa}} + \mathcal{M}_{t_{be}}$ | $\mathcal{M}_{s_{fa}} + \mathcal{M}_{t_{el}}$ | $\mathcal{M}_{s_{fa}} + \mathcal{M}_{t_{gr}}$ | Avg |
| Ours (transfer) | 0.71 | 0.70 | 0.69 | **0.70** |
| Ours (transfer) | 0.41 | 0.39 | 0.40 | **0.40** |

Table 3: Transfer performance under **different source tasks**. For each method, the first row reports the mean scores, and the second row reports the success rates. Task abbreviations: **fa** = fashion, **be** = beauty, **el** = electronics, **gr** = grocery.

| Method | $\mathcal{M}_{t_{be}}$ | $\mathcal{M}_{t_{be}}$ | $\mathcal{M}_{t_{be}}$ | Avg |
|---|---|---|---|---|
| LLM only | 0.55 | - | - | 0.55 |
| LLM only | 0.28 | - | - | 0.28 |
| RMMBR.(w/o transfer) | 0.68 | - | - | 0.68 |
| RMMBR.(w/o transfer) | 0.38 | - | - | 0.38 |
| | $\mathcal{M}_{s_{fa}} + \mathcal{M}_{t_{be}}$ | $\mathcal{M}_{s_{el}} + \mathcal{M}_{t_{be}}$ | $\mathcal{M}_{s_{gr}} + \mathcal{M}_{t_{be}}$ | Avg |
| Ours (transfer) | 0.70 | 0.72 | 0.71 | **0.71** |
| Ours (transfer) | 0.40 | 0.41 | 0.40 | **0.40** |

Tables 2 and 3 present cross-task transfer results across four product domains: *fashion* (fa), *beauty* (be), *electronics* (el), and *grocery* (gr). We examine two complementary transfer settings. Table 2 varies the *target task* while fixing the source memory, evaluating the agent's ability to adapt to unseen domains using knowledge from a single source. Table 3 varies the *source task* while keeping the target fixed, testing robustness when transferable knowledge is drawn from heterogeneous sources. In both cases, our method surpasses all baselines, achieving consistently higher scores and success rates. The observed improvements stem from the interaction of two key components. First, the *source experience memory* enables the agent to reuse transferable strategies across domains, reducing the need for extensive task-specific exploration. This explains why adaptation remains effective even when source and target domains are semantically distant. Second, the *dynamic retrieval mechanism* regulates how source and target memories are weighted during inference, allowing the agent to selectively transfer useful patterns while avoiding negative transfer. This adaptive balance underlies the strong generalization observed in both transfer directions, demonstrating that our approach can efficiently exploit prior knowledge while preserving target-task specialization.

## 4.3 ABLATION STUDY

### 4.3.1 EFFECT OF SUPPRESSION THRESHOLD $n$.

As detailed in Section *Dynamic Memory Retrieval Mechanism*, the parameter $n$ sets a minimum coverage threshold that determines when to suppress retrieval from source memory $\mathcal{M}_s$, thereby shifting focus to target memory $\mathcal{M}_t$ once sufficient task-specific experience is available. This design improves retrieval efficiency and mitigates negative transfer. We evaluate $n \in 1, 2, 3, 4$ to capture different adaptation stages: smaller $n$ accelerates reliance on $\mathcal{M}_t$, while larger $n$ prolongs dependence on $\mathcal{M}_s$. As shown in Table 4, $n = 2$ yields the best performance (0.70 average score,

Table 4: Performance under different memory transition thresholds $n$.

| Setting | Avg Score | Success Rate |
|---|---|---|
| $n = 1$ | 0.67 | 0.38 |
| $n = 2$ | **0.70** | **0.40** |
| $n = 3$ | 0.71 | 0.39 |
| $n = 4$ | 0.71 | 0.38 |

0.40 success rate). A smaller value ($n = 1$) detaches from $\mathcal{M}_s$ too early, reducing performance (0.67 / 0.38), while larger values ($n = 3, 4$) delay suppression and introduce noise, leading to marginal or degraded results. Overall, a moderate threshold ($n = 2$) achieves the best balance between transfer and adaptation.

### 4.3.2 EFFECT OF TRANSFER AND DYNAMIC MEMORY RETRIEVAL.

Table 5: Performance comparison across ablation settings.

| Setting | Avg Score | Success Rate |
|---|---|---|
| RMMBR. (w/o transfer) | 0.68 | 0.38 |
| Ours (w/o DMRM) | 0.69 | 0.38 |
| Ours (Full DMRM) | **0.70** | **0.40** |

To further isolate the contributions of transfer learning and the proposed retrieval mechanism, we compare three settings: (i) **No Transfer**, where the agent relies solely on target memory $\mathcal{M}_t$ without access to source experiences; (ii) **w/o DMRM**, where both source and target memories are available but retrieval follows a static strategy without adaptive suppression; and (iii) **Full DMRM**, our proposed approach with dynamic retrieval. As shown in Table 5, No Transfer yields the weakest performance (average score: 0.68, success rate: 0.38), indicating that access to prior experiences is crucial for enhancing target-task learning. Adding source memory without adaptive regulation (w/o DMRM) provides only marginal improvement (0.69 / 0.38), as indiscriminate retrieval introduces noise and negative transfer that offset potential gains. In contrast, Full DMRM achieves the best performance (0.70 / 0.40). This improvement arises from the *dynamic retrieval mechanism*, which adaptively balances the use of source and target experiences, amplifying beneficial transfer while suppressing irrelevant knowledge. These results confirm that both components are indispensable: cross-task memory supplies transferable knowledge, while dynamic retrieval ensures its effective and selective integration.

## 5 CONCLUSION

In this work, we have developed a memory-augmented LLM-based agent with cross-task experience learning to enhance adaptability and generalization in multi-turn decision-making. Our design has extended conventional task-specific memory with a source experience memory that stores transferable knowledge from related but distinct tasks, and has introduced a Dynamic Memory Retrieval Mechanism that adaptively balances task-specific and cross-task experiences throughout interaction. We have extensively evaluated our approach on the WebShop benchmark, where it has consistently outperformed strong memory-augmented baselines in both task success rate and generalization. These findings have demonstrated that integrating transferable memory with adaptive retrieval not only has improved data efficiency but also has enabled more robust and flexible decision-making in complex, multi-turn environments.

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

## A    APPENDIX

## B    ETHICS STATEMENT

This work complies with the ICLR Code of Ethics. Our study does not involve human participants or animal experimentation. All datasets employed are publicly available and used strictly in accordance with their licensing and usage guidelines, thereby avoiding any violation of privacy or intellectual property. We have carefully considered potential risks of bias or unintended harm. No personally identifiable information was used, and no experiments were conducted that could raise privacy, fairness, or security concerns. We are committed to conducting and presenting our research with transparency, integrity, and respect for ethical standards.

## C    LLM USAGE

Large Language Models (LLMs) were used solely to assist in the preparation of this manuscript. Their contributions were limited to language refinement tasks such as improving readability, enhancing clarity, and ensuring stylistic consistency with academic writing standards. The LLM was not involved in conceptual development, experimental design, implementation, or analysis. All scientific ideas, methodologies, and results were conceived and validated independently by the authors. The authors take full responsibility for the final content of this paper. We have carefully reviewed all text produced with LLM assistance to ensure accuracy, originality, and compliance with ethical guidelines, thereby avoiding plagiarism or scientific misconduct. The LLM was employed strictly as a writing aid, without influencing the research substance.

