# OpenReview forum: "Memory-Augmented Large Language Model-Based Agent with Cross-Task Experience Learning"
_ICLR.cc/2026/Conference — Submitted to ICLR 2026_

### Official Review · Reviewer_nXBR · 2025-10-15

**Soundness:** 2
**Presentation:** 3
**Contribution:** 2
**Rating:** 4
**Confidence:** 4

**Summary:**

This paper addresses two critical limitations of existing memory-augmented Large Language Model (LLM)-based agents: low data efficiency (relying on extensive task-specific interaction data for early training) and poor adaptability (using static memory retrieval strategies that fail to balance cross-task knowledge and current task needs) . To resolve these, it proposes a memory-augmented LLM agent with cross-task experience learning, centered on a dual-memory architecture and a dynamic retrieval mechanism, validated on the WebShop benchmark (a multi-turn online shopping simulation with 1M+ product descriptions and 12k human instructions)

**Strengths:**

- Prior memory-augmented agents (e.g., REMEMBERER, Reflexion) relied solely on task-specific self-derived experience (e.g., only current shopping interactions for WebShop), leading to low data efficiency: they required extensive early training data to compensate for knowledge gaps. The paper breaks this constraint by proposing a dual-memory architecture.

- Existing methods (e.g., REMEMBERER) used fixed retrieval rules (e.g., always retrieve 2 experiences) that failed to adapt to task progression, leading to negative transfer (irrelevant cross-task knowledge) or redundancy. The paper’s Dynamic Memory Retrieval Mechanism (DMRM) addresses this by balancing similarity-based ranking (instruction similarity via SentenceTransformer + observation similarity via WebShop page type matching) with token budget control (avoiding LLM context overflow).

**Weaknesses:**

- Over-Reliance on WebShop Limits Generalizability: The paper exclusively validates its framework on the WebShop benchmark but fails to test diverse multi-turn decision scenarios—a critical gap that undermines the claim of "broad adaptability" and restricts the framework’s practical applicability.

- The paper’s source memory (\(M_s\)) relies on manual curation of "related tasks" and "high-utility experiences" (e.g., selecting fashion shopping as a source for electronics shopping, filtering only reward=1.0 experiences) but provides no automated mechanisms—creating high labor costs and limiting scalability to new tasks

- The paper’s Dynamic Memory Retrieval Mechanism (DMRM) relies on shallow similarity metrics (cosine similarity for instructions, page type matching for observations) that fail to capture nuanced semantic relationships—leading to suboptimal retrieval of cross-task knowledge and potential negative transfer

- The target memory (\(M_t\)) is designed to "incrementally append" new experiences without compression or forgetting mechanisms—leading to exponential growth in memory size over long-term deployment, which degrades retrieval speed and introduces redundant information

**Questions:**

see weakness

---

### Official Review · Reviewer_RXKz · 2025-10-19

**Soundness:** 2
**Presentation:** 3
**Contribution:** 2
**Rating:** 4
**Confidence:** 4

**Summary:**

The paper introduces a memory-augmented LLM-based agent that incorporates cross-task experience learning to improve data efficiency and adaptability. It features a dual-memory architecture with Target memory for task-specific experiences，Source memory for transferable knowledge from related tasks. A dynamic retrieval mechanism adaptively balances these two memories based on interaction context and task progression. Evaluated on the WebShop benchmark, the agent outperforms strong baselines in task success rate and cross-domain generalization.

**Strengths:**

* The cross-task memory design and dynamic retrieval mechanism (DMRM) enable transferable knowledge reuse across tasks, offering a fresh and sound approach to improving sample efficiency and generalization.
* Experiments on the multi-turn WebShop benchmark demonstrate consistent gains in task success rate and cross-domain generalization, validating the method’s effectiveness  to a certain extent.
* The paper is well-written and logically structured, making the approach easy to follow and replicate.

**Weaknesses:**

* While the proposed method demonstrates improved cross-task learning within WebShop, its evaluation is limited to a single environment. All WebShop tasks operate under the same structured API and UI; differences across tasks lie mainly in textual product descriptions. This undermines the claim of generalization and cross-task transferability.

* The paper emphasizes reducing early-stage interaction cost via cross-task memory. However, no quantitative evidence is presented for data efficiency. The authors should add plots or metrics to validate whether memory-based transfer truly leads to faster or cheaper learning.

**Questions:**

1. The current work is evaluated only on WebShop. It is recommended to conduct additional experiments on more heterogeneous web environments (e.g., Mind2Web, WebArena) to verify its generalization and cross-task transferability.
2. Does memory truly reduce early-stage interaction cost? Provide success-over-time curves (success rate vs. episodes/steps) or other metrics comparing your method to baselines, not just final success.

---

### Official Review · Reviewer_irPg · 2025-10-28

**Soundness:** 2
**Presentation:** 2
**Contribution:** 2
**Rating:** 2
**Confidence:** 4

**Summary:**

This paper proposes a memory-augmented LLM-based agent with cross-task experience learning to address two limitations of existing memory-augmented agents: (1) low data efficiency due to reliance on extensive task-specific interaction data, and (2) static memory retrieval strategies that hinder adaptability. The method introduces a dual-memory architecture:

Source Experience Memory (Ms): Stores transferable knowledge from related tasks.
Target Experience Memory (Mt): Accumulates task-specific experiences during interactions.

A Dynamic Memory Retrieval Mechanism (DMRM) adaptively balances retrieval from Ms and Mt based on task progression, mitigating negative transfer. Evaluated on the WebShop benchmark demonstrates improved data efficiency and generalization.

**Strengths:**

(1)Effectively addresses limitations of memory-augmented LLM agents (data inefficiency, static retrieval) with a well-motivated solution.

(2)The dual-memory design (Ms + Mt) and DMRM sound feasible, enabling adaptive knowledge transfer while preserving task-specific specialization.

(3)Relatively complete experimental evaluation. Extensive experiments on WebShop include:
Cross-task transfer across domains (fashion, beauty, electronics, grocery).
Ablation studies on suppression threshold n and DMRM components.

**Weaknesses:**

(1)The paper proposes some concepts and ideas, but the implementation details are not clearly described. The paper also does not provide sufficient analysis and demonstration of the rationality of these concepts and ideas.

(2)Some descriptions in the paper are unclear or contradictory. For example, regarding the selection of topk in Formula 3, the paper mentions that the agent ranks candidate entries from Ms ∪ Mt using contextual similarity. If the selection is based on Ms ∪ Mt, how can we achieve a balance and give higher priority to Mt?

(3)Some of the methods proposed in the paper are simple but not necessarily sound. For example, in the adaptive retrieval process, the paper mentions that the number of retrieved entries k is dynamically adjusted according to the model's context window. Existing LLMs can have very large context windows, and adjusting k based on the maximum context window is neither economical nor a good practice. Even within the permissible range of the LLM's context window, providing the LLM with more context is not necessarily better.

(4）There is no introduction to the implementation of the experiment, nor any relevant supplementary materials. Considering the incomplete description of details in the previous theoretical part, it is not clear as a whole how the work of the paper is implemented and why it is effective.

**Questions:**

(1)How are "transferable" experiences in Ms identified and filtered? Could noisy or irrelevant experiences degrade performance.

(2) (line 288) The paper states that "once Mt contains at least n high-quality successful trajectories, retrieval from Ms is reduced or disabled." The question is how to identify high-quality successful trajectories?

(3) (line 443) The ablation experiment section of the paper mentions that "a moderate threshold (n = 2) achieves the best balance between transfer and adaptation." The question is, with such a small value of n, will Ms cease to function very early, and how can transferable knowledge be utilized then?

(4)Why should Ms be set to static, and what kind of negative transfer will Ms cause.

(5)Why not compare it with some recent, stronger baselines in memory-augmented aspects?

(6)Beyond WebShop, has the method been tested in non-e-commerce settings.

---

### Official Review · Reviewer_ZV6F · 2025-10-30

**Soundness:** 2
**Presentation:** 3
**Contribution:** 3
**Rating:** 2
**Confidence:** 4

**Summary:**

The paper proposes a memory-augmented LLM agent that introduces a cross-task experience learning mechanism, allowing the agent to reuse knowledge from previously completed tasks while dynamically adapting to new ones. It also adds a Dynamic Memory Retrieval Mechanism (DMRM) to balance between task-specific and cross-task experiences.

**Strengths:**

1. The paper identifies two well-known limitations: (a) over-reliance on task-specific experience, and (b) static retrieval strategies.

2. WebShop is an appropriate benchmark for multi-turn decision-making with clear metrics. Comparison to well-known baselines like ReAct and REMEMBERER is relevant.

**Weaknesses:**

1. **Lack of conceptual novelty**: The proposed cross-task memory is essentially a pre-filled experience buffer, not a fundamentally new paradigm. Prior works such as REMEMBERER, Reflexion, and GITM have already explored experience-based reuse and memory replay. The `dynamic retrieval` is the claimed benefits over REMEMBERER but it is essentially an adaptive weighting based on the number of successful trajectories, with no learned mechanism or new retrieval objective.

2. **Weak Empirical Validation**: (a) Reported gains are marginal: e.g., success rate 0.40 vs. 0.38 for REMEMBERER — an absolute improvement of only 2%. This is within noise for WebShop, which is highly stochastic. (b) The authors do not show sample efficiency curves, so the claim of improved data efficiency is unsupported. (c) The ablation study is minimal: only threshold n and DMRM are tested. No analysis on retrieval size k, memory size, or domain similarity.

3. **Limited Evaluation Scope**: The entire study relies on one benchmark (WebShop). There is no test on other environments such as ALFWorld, MiniWoB, or BabyAI, which limits generality. Also, “Cross-task” is narrowly defined as different product domains (e.g., fashion → electronics), which share a similar interface and structure — thus not demonstrating true cross-domain generalization.

4.  **Lack of comparison to stronger baselines**: The baselines are outdated (ReAct, REMEMBERER). There’s no comparison to recent Memory-Augmented Agents such as MemGPT, Mem0, Zep, MIRIX, MEM1. The results are not competitive with state-of-the-art long-context models (such as GPT-4.1 which has 1M context window), which might outperform this method simply by concatenating history.

**Questions:**

See weaknesses.

**Details Of Ethics Concerns:**

I don't have any ethics concerns.

---

### Meta-Review · Area_Chair_Bj5C · 2025-12-14

**Summary:**

The dual-memory design and adaptive retrieval mechanism are viewed as reasonable extensions of existing memory-based agent frameworks. However, the claimed improvements over prior work (e.g., REMEMBERER) are marginal and lack rigorous ablation studies or statistical significance testing. The evaluation is limited in scope, as all experiments are conducted solely on the WebShop benchmark, which constrains the generalizability of the results. Additionally, the retrieval mechanism relies on heuristic similarity metrics and does not incorporate learning-based or principled optimization. Incorporating comparisons with more recent and stronger baselines, such as MemGPT or Mem0, would significantly strengthen the empirical contribution and better highlight the method’s potential.

**Reviewer Concerns:**

No author response was submitted.

**Reviewer Scores:**

No author response was submitted.

---

### Decision · Program_Chairs · 2026-01-26

Reject